# A Systematic Review of Areal Units and Adjacency Used in Bayesian Spatial and Spatio-Temporal Conditional Autoregressive Models in Health Research

**DOI:** 10.3390/ijerph20136277

**Published:** 2023-07-01

**Authors:** Zemenu Tadesse Tessema, Getayeneh Antehunegn Tesema, Susannah Ahern, Arul Earnest

**Affiliations:** 1School of Public Health and Preventive Medicine, Monash University, Melbourne, VIC 3004, Australia; 2Department of Epidemiology and Biostatistics, Institute of Public Health, College of Medicine and Health Sciences, University of Gondar, Gondar P.O. Box 196, Ethiopia

**Keywords:** Bayesian, spatio-temporal, adjacency matrices, conditional autoregressive model

## Abstract

Advancements in Bayesian spatial and spatio-temporal modelling have been observed in recent years. Despite this, there are unresolved issues about the choice of appropriate spatial unit and adjacency matrix in disease mapping. There is limited systematic review evidence on this topic. This review aimed to address these problems. We searched seven databases to find published articles on this topic. A modified quality assessment tool was used to assess the quality of studies. A total of 52 studies were included, of which 26 (50.0%) were on infectious diseases, 10 (19.2%) on chronic diseases, 8 (15.5%) on maternal and child health, and 8 (15.5%) on other health-related outcomes. Only 6 studies reported the reasons for using the specified spatial unit, 8 (15.3%) studies conducted sensitivity analysis for prior selection, and 39 (75%) of the studies used Queen contiguity adjacency. This review highlights existing variation and limitations in the specification of Bayesian spatial and spatio-temporal models used in health research. We found that majority of the studies failed to report the rationale for the choice of spatial units, perform sensitivity analyses on the priors, or evaluate the choice of neighbourhood adjacency, all of which can potentially affect findings in their studies.

## 1. Introduction

The advancement of geographic information systems in medicine and public health has led to an increase in spatial modelling, which is an important component of spatial epidemiology that analyses georeferenced data on health and health-related outcomes [1,2]. This growth is primarily due to improvements in the availability of data with geographic information and advancements in statistics, particularly in computing speed, which enables researchers to apply advanced spatial and spatio-temporal models that consider spatial, and both space and time dimensions, respectively [3,4].

Small-area-level analysis, also known as disease mapping, is a key component of spatial epidemiology [5]. It involves the use of georeferenced data to create maps that show the spatial distribution of health outcomes or diseases at a fine-grain level, such as a neighbourhood or census tract. By analysing these maps, researchers can identify patterns and hotspots of disease, as well as explore the relationship between health outcomes and environmental factors such as air pollution, water quality, and access to healthcare [6]. This approach has become increasingly important in public health research, as it can guide policy decisions and interventions aimed at improving population health [7].

Disease mapping in regional areas with low population counts is often limited by data sparseness. To address this challenge, the Basag, York, and Mollie (BYM) model was developed in 1991, which allows borrowing of strength in outcomes from nearby locations [8]. The conditional autoregressive (CAR) model is often applied in small-area ecological studies to map outcome measures to an area level and to identify relationships with covariates [9,10].

This modelling technique is often used to describe the spatial variation of quantities of interest in the form of summaries or aggregates over subregions [11]. The CAR model random effect component contains both spatially structured and unstructured spatial random effects known as convolution prior [12]. Since different geographical areas generally have similar environmental conditions if they are neighbouring areas, the CAR model borrows information from the neighbours that share a common boundary (neighbour) for each areal unit in the data analysis [13]. There are four variants of CAR priors: intrinsic, convolution, Cressie, and Leroux [14].

The Modifiable Area Unit Problem (MAUP) is a statistical concern that occurs with the aggregation of spatial data by modifying the shape, size, and/or the orientation of spatial locations in geographic areas [15]. This problem has two interrelated effects: firstly, grouping the same data into larger geographic areas can lead to different inferences, and secondly, variability in results may occur due to different formulations of the areas [16]. Research conducted by Mei Ruu Kok et al. [17] found that estimates obtained from different spatial scales can vary significantly. When analysing area-level data, it is necessary to determine the appropriate spatial scale of analysis beforehand. A study conducted by Hanigan et al. [18] showed that the spatial scale had a significant impact on statistical inference and concluded that the appropriate spatial scale should be identified for spatial analysis.

The effectiveness of the smoothing properties of the CAR model relies on the types of adjacency matrices or neighbours that are specified. Therefore, prior to risk mapping or ecological modelling, an exploratory analysis of neighbourhood weight should be developed. Earnest et al. conducted studies that demonstrated significant differences in the smoothing properties of CAR models depending on the selection of adjacency matrices [13]. In addition, a study conducted by Duncan et al. [19] showed that the selection of the adjacency weight matrix can have an immense effect on model fit and inference.

Bayesian spatial and spatio-temporal modelling is a statistical technique used to analyse and model data that have both spatial (geographical) and temporal components [20]. These models are based on the principles of Bayesian statistics, which provide a framework for incorporating prior knowledge and uncertainty into statistical inference. In the Bayesian framework, the posterior distribution is made up of both the prior and the data [21]. In spatial modelling studies, most published articles have utilized non-informative priors to simplify model formulation [22]. However, an important aspect of Bayesian spatial analysis is determining the hyperprior distribution of the variance parameter (precision).

There have been three systematic reviews examining the use of Bayesian spatial and spatio-temporal modelling to create risk maps in cancer, dengue, and public health research. Wah et al. [23] conducted a review on Bayesian spatio-temporal models for cancer incidence, Aswi et al. [24] reviewed Bayesian spatial and spatio-temporal approaches for modelling dengue fever, and Byun et al. [25] reviewed spatial and spatio-temporal analyses in public health research in Korea. However, our review is not limited to a specific disease or geographic region as per previous reviews. Rather, we focus on the selection of spatial adjacencies, areal unit, and the justification for prior selection in Bayesian spatial and spatio-temporal modelling. Despite these reviews that have been undertaken, there are still unresolved issues around the selection of the appropriate areal unit of analysis and the choice of the adjacency. Our review aimed to further consider this problem.

## 2. Materials and Methods

### 2.1. Data Source and Search Strategy

This methodological systematic review adhered to the Preferred Reporting Items for Systematic Reviews and Meta-Analysis (PRISMA) checklist [26]. The PRISMA checklist aids systematic reviewers in providing transparent reporting by assisting them in clearly stating the purpose of the review, describing the authors’ actions and methods, and presenting the findings obtained [27]. The focus of this systematic review is on peer-reviewed health and health-related research that employed Bayesian spatial and spatio-temporal conditional autoregressive models and reported the selection of the spatial unit, adjacency matrix, or priors. We registered this review with the PROSPERO international prospective register of systematic reviews under the registration number CRD42022371293. Our search was comprehensive and included studies from 2012 onwards that used Bayesian spatial and spatio-temporal conditional autoregressive models. Studies also had to have reported the selection of the areal unit or adjacency matrix or priors for any health or health-related outcomes, with no geographic restrictions.

We conducted a thorough search of seven databases, namely Web of Science, PubMed, MEDLINE, Scopus, PsycINFO, Emcare, and Embase, to identify relevant studies. We also searched Google Scholar to uncover any additional studies. The search was conducted on September 8, 2022, using specific search terms for Bayesian spatial and spatio-temporal studies which at least reported areal unit, adjacency matrix, or prior selection (Appendix A). For this review, we considered studies published between 1 January 2012 and 8 September 2022, with no geographic restrictions.

The articles retrieved from each database were combined into a single file using Endnote software. This file was then imported into Covidence for further screening. During the title, abstract screening, and full-text screening stages, duplicates were identified and removed both manually and automatically using Endnote and Covidence. The search was conducted using a combination of keywords including “Bayesian”, “spatial”, “spatialtemporal”, “Spatial-temporal”, “spatiotemporal”, “spatio-temporal”, “space-time”, “geo-temporal”, “geotemporal”, “geographic-temporal”, and “conditional autoregressive”, as outlined in Appendix A.

### 2.2. Inclusion and Exclusion Criteria

This systematic review focused on peer-reviewed articles published in English between January 2012 and September 2022. Earlier articles were excluded as they may have used outdated methodology and techniques. The review included articles that reported the use of Bayesian spatial and spatio-temporal conditional autoregressive models and with information on the selection of areal unit, adjacency matrix, and priors. Two authors (ZTT and GAT) independently screened titles, abstracts, and full texts for eligibility, with conflicts resolved through discussion with a third reviewer (AE). The review had no geographic restrictions and included all health and health-related outcomes. Methodological reviews, conference abstracts, studies in languages other than English, cluster detection studies, animal studies, and non-Bayesian studies were excluded.

### 2.3. Data Extraction

To facilitate data extraction, a standardized template was developed in Microsoft Excel that incorporated relevant information based on the review questions. This template included bibliographic details, research objectives, data sources, research category, types of adjacencies used, available spatial scales of the study country, covariate types used, data analysis methods, modelling approaches, generated results, identified methodological gaps, and potential future research directions.

### 2.4. Risk of Bias Assessment

Two authors (ZTT and GAT) independently assessed the methodological quality of the included studies in this systematic review. We used an eight-point scoring system that was updated and modified to evaluate each study’s quality based on its aim and objective, model validity, and overall results. The standardised item list was employed to grade all included studies to determine their quality and risk of bias [28,29]. The bias assessment tool consists of eight questions with possible answers ranging from zero to two, and the maximum score is 16. Based on the total score, the overall quality level was classified as low (score less than 8), medium (score 8–10), high (score 11–13), or very high (score >13).

### 2.5. Data Synthesis and Analysis

Microsoft Excel and STATA version 17 software were used for data entry and analysis, respectively. The results of the included studies were synthesized and presented in the form of text, tables, and figures. Summary statistics, such as proportions, means, medians, and ranges, were used to describe the findings of the studies.

## 3. Results

### 3.1. Characteristics of Included Studies

The search for relevant articles using seven databases and manual search in Google and Google Scholar resulted in 1535 published articles. After removing 819 duplicates using Covidence and Endnote software, 716 articles were eligible for title and abstract screening. Of these, 419 articles were excluded based on title screening and 165 articles were excluded based on abstract screening. After full-text review of 132 articles, 80 articles were excluded, leaving 52 studies that met the eligibility criteria for this systematic review. The details of the excluded and included studies can be found in Appendix A and Figure 1, respectively.

### 3.2. Publications by Year and Country of Study

This review encompasses studies that have been published since 8 September 2012. The number of publications experienced a significant increase after 2017, peaking in 2020. A considerable proportion of studies, almost two-thirds (*n* = 33, 63.4%), were published after 2015. From this review, Australia (*n* = 11, 21.15%) and Indonesia (*n* = 6, 11.5%) were identified as countries with a high number of publications (refer to Figure 2 and Figure 3).

### 3.3. Summary Statistics of Included Studies

Of the total of 52 studies, 26 (50.0%) were on infectious diseases, 11 (21.1%) on malaria [30,31,32,33,34,35,36,37,38,39], 8 (15.3%) on Dengue fever [40,41,42,43,44,45], 4 (7.6%) on HIV [46,47,48,49], and 3 (5.7%) on TB [50,51,52]. Overall, 10 (19.2%) studies were on chronic diseases [53,54,55,56,57,58,59,60,61,62], 8 (15.5%) on maternal and child health [63,64,65,66], 3 (5.7%) on nutrition [67,68,69] and 5 (9.6%) were applied to other health-related outcomes [58,70,71,72,73]. Most of the studies (42, 80.7%) used an ecological study design [30,31,33,34,37,38,39,40,41,42,43,44,45,46,47,51,52,53,54,55,56,58,59,60,61,62,64,65,66,67,69,71,72,73,74,75,76,77,78,79,80,81]. Twenty-four studies (46.1%) used data from a national health survey or Demographic and Health Survey (DHS), while twenty other studies (38.4%) used data from hospital/administrative records. The studies included in this review used different spatial units for mapping. The number of spatial units in the reviewed studies ranged from a minimum of 10 to a maximum of 1325. About 9 (17.3%), 7 (13.1%), 6 (11.5%), and 6 (11.5%) studies’ spatial unit of analysis were districts [37,44,46,56,62,68,74,81], counties [32,34,48,49,52,75,77,78], regions [30,31,38,59,72,80], and provinces [35,41,45,50,51,53], respectively. The majority of the studies (80.7%) used years as a temporal unit measure and 9 (17.3%) studies used months as a temporal unit measure. A total of 46 (88.5%) studies used the MCMC estimation technique for their Bayesian modelling [31,32,33,35,36,37,38,39,40,41,42,43,44,45,46,47,48,49,50,51,52,54,55,57,58,59,60,61,62,63,64,65,68,69,70,71,72,73,74,75,76,77,78,79,80,81] and 6 (11.5%) studies used INLA as their estimation technique [30,34,53,56,66,67].

Different statistical software was used for modelling in the reviewed studies. Of these, 31 (59.6%) used WinBUGs, followed by 11 studies which used R software (21.1%). A total of 30 (57.7%) studies used the Bayesian spatial Poisson regression model, 16 (30.7%) used Bernoulli/Binominal modelling, and 6 (11.5%) used Negative Binomial modelling. Different types of covariates were included in the reviewed studies. From a total of 52 studies, 21 (41.1%) studies included demographic variables, 19 (37.2%) studies included socio-economic variables, and 16 (30.7%) studies included climatology/environmental variables in their model development. Of the 52 studies, 24 (46.1%) reported model diagnostics in their study. The majority of the studies (44, 84.6%) reported their data in a choropleth map [30,31,32,33,34,35,36,37,38,39,40,41,42,44,45,46,47,48,49,50,51,52,53,55,56,57,58,59,62,63,64,65,67,68,69,70,71,73,74,75,76,77,78,79,80], and only 5 (9.6%) studies provided script codes as a Appendix A [41,50,51,71,72]. A total of 28 (53.8%) studies reported that burn-in, thinning, and iteration were used in their Bayesian modelling. Of a total of 52 studies, only 2 studies reported the reason for the use of their specified adjacency matrix [63,65] (Table 1).

### 3.4. Prior Distribution Selection, Sensitivity Analysis, and Adjacency Matrices

Of the 52 studies, 19 (36.5%) assigned flat prior distribution for the intercept [31,38,39,40,43,44,50,51,57,60,69,70,71,72,73,74,77,80,81] and 17 (32.6%) studies did not report any prior distribution for the intercept [30,33,34,36,47,53,56,58,59,61,63,64,65,66,68,75,79]. For the regression coefficients, 36 (69.2%) studies used normal distribution for prior distribution and 11 (21.1%) articles did not report the types of prior distribution used for the regression coefficient. For the spatial structured and unstructured random component of the model, 48 (92.3%) studies reported prior distribution of the precision term. Of these, 31 (59.6%) studies reported the specific shape and scale parameter value of inverse gamma distribution for their models. From this review, only 8 (15.3%) studies [41,42,51,54,55,63,71,78] conducted sensitivity analysis for hyperprior distribution of the precision term. Different types of adjacency matrices were identified in this review. A total of 39 (75%) articles used Queen contiguity adjacency matrix for smoothing spatially structured random effects in their modelling. Eight articles in the included studies did not report the types of adjacency matrices used in their modelling (Table 2).

### 3.5. Modifiable Area Unit Problem (MAUP)

For this systematic review, the countries of the included articles had different spatial scales available for the analysis as described in Table 3. From a total of 52 articles, only 6 (11.5%) [55,63,66,70,73,79] studies reported the reasons for use of the specified spatial scale for analysis. The studies conducted by Chou et al. [70] mentioned that they used postcodes for spatial analysis because it was the only available spatial scale in the dataset. A study conducted by Li et al. [63] stated that they used the SA3 spatial scale because the objective of the study was spatial mapping at the regional level in Australia. The studies conducted by Qi et al. [73] noted that they used LGA because the LGA boundaries provide a more stable population than SLA boundaries (Table 3).

### 3.6. Key Considerations in Applying Bayesian Spatial and Spatio-Temporal Conditional Autoregressive Models and Methodological Gaps

Bayesian modelling requires researchers to be mindful of several methodological factors, such as appropriate prior selection, adjacency/neighbourhood matrices, and the areal unit of analysis. Neglecting these factors can significantly affect the results and the interpretation of data. Therefore, it is crucial for researchers to carefully consider these factors in their model development.

The findings of this systematic review indicate that only a limited number of articles included information about the choice of prior selection, adjacency/neighbourhood matrices, and areal unit of analysis in their Bayesian spatial and spatio-temporal modelling. The justifications for the selection of prior distributions, neighbourhood structures, and spatial unit used in mapping studies were found to be diverse and varied.

Among a total of 52 articles, only 8 (15.39%) of them [41,42,51,54,55,63,71,78] conducted sensitivity analysis for the precision term of spatial structured and unstructured random effect and justified their prior distribution selection. Aswi et al. [41] reasoned that they selected prior distribution when Bayesian inference was robust and not sensitive to the choice of prior, as summarized in Table 4. From a total of 52 studies, only 2 [63,65] reported the reason of selection of adjacency matrix. Lome-Hurtado et al. [65] selected Queen contiguity because the use of a distance-based model was stated to be more complex and created unrealistic spatial dependence. Studies performed by Li et al. [63] used the Queen method of adjacency given that there were a number of areas with multiple neighbours, as well as stating that the use of a distance-based method would introduce model complexity and was unrealistic.

Of the 52 studies, 6 (11.53%) reported the reason for the choice of the spatial scale used for mapping [55,63,70,71,73,79]. These studies justified their choice of spatial scale based on data availability, study objectives, and location stability. Of the 52 studies, only 8 identified methodological gaps and future research directions [40,41,51,56,65,71,74,75]. The study conducted by Darikwa et al. [56] justified that large geographical units of analysis may mask some information of interest. The efficiency may be improved by using smaller units of analysis. Another study conducted by Akter et al. [40] noted that the impact of using different smoothing priors on inference was not investigated in their study and recommended investigation for future research (Table 4).

### 3.7. Assessment of Quality

The risk of bias tool for assessment was adapted from quality assessment tool for modelling studies from Fone et al. (2003) and Harris et al. (2016) [28,29]. The tool has three broad criteria: the screening questions, the validity of the model questions, and the overall result and study conclusion questions. The screening criteria include two questions (Does the paper clearly address aims and objectives? and Is the setting and population clearly defined?). The validity of the model includes four questions (Is the model structure clearly described and appropriate for the research question? Are the modelling methods appropriate for the research question? Are the parameters, ranges and data source specified? and Is the quality of data considered?). The overall result and study conclusions comprise two questions (Have the results been clearly and completely presented? Additionally, are the results appropriately interpreted and discussed in context?). The studies included in this review were evaluated using the above-listed eight questions with possible answers ranging from zero to two (0, none; 1, poor; 2, good) and a maximum score of sixteen. Each reviewed study was categorized as either very high quality (score > 13), high quality (score 11–13), medium quality (score of 8–10), or low quality (score < 8). The median quality score for this review was 12, with the minimum score being 8 and the maximum score being 16 (Appendix A).

## 4. Discussion

This systematic review aimed to examine the utilization of Bayesian spatial and spatio-temporal conditional autoregressive models in health and health-related research, with a specific focus on the selection of spatial adjacencies, areal units, and the justification of priors used in the studies. A total of 52 articles were included in this review, all of which utilized Bayesian spatial and spatio-temporal conditional autoregressive modelling and reported information on adjacency matrices, areal units, and priors in their investigation of health and health-related outcomes.

The majority of the conditional autoregressive (CAR) models in this systematic review were applied to infectious diseases, with chronic diseases being the second most common focus. Infectious diseases are a significant global cause of morbidity and mortality, including the current Coronavirus-2 pandemic [82,83], with 50% of the reviewed papers relating to infectious diseases. Due to their mode of transmission, infectious diseases have the potential to spread to nearby areas and populations, particularly respiratory communicable diseases like SARS, COVID-19, Ebola, and measles [49,84,85,86]. The Bayesian modelling framework is well suited for incorporating prior information from previous disease strains, and it aligns with Tobler’s first law of geography, which suggests that “everything is related to everything else, but near things are more related than distant things” and emphasizes spatial dependence [87]. Hence, CAR models are highly relevant to the analysis of infectious diseases.

Of the 52 studies, 31 (59.6%) reported relative risk as the effect measure, followed by the mean coefficient of 12 (23.0%). A considerable number of the studies included in the analysis reported smoothed relative risk as the preferred measure of effect size. This preference can be attributed to the fact that many of the studies utilized an ecological study design, for which relative risk is considered an appropriate measure of effect size.

Out of the 52 studies reviewed, 42 (80.7%) employed an ecological study design. This indicates that ecological study design is the most commonly utilized approach for disease mapping at a more refined geographic level, as it can aid planners and decision-makers in allocating resources effectively [12]. Unlike other study designs, ecological study design can be easily generated from various sources of data and reports, ranging from individual/patient-level data to group-level data at any geographic level. Of the studies reviewed, the primary data sources included DHS survey datasets for 24 (46.1%) studies and hospital/administrative dataset survey data for 20 (38.4%) studies. This may be due to the fact that survey datasets frequently include location data that contain longitude and latitude measurements for each enumeration area and provide geographic-related covariates within the survey dataset [87].

Out of the 52 studies reviewed, 46 (88.5%) used Markov Chain Monte Carlo (MCMC) for model fitting and inference in their Bayesian spatial and spatio-temporal modelling. The main reason that most scholars use MCMC is that it provides a flexible framework for sampling from complex probability distributions without dealing with the intractability of complex equations (e.g., conditional risks between adjacent regions). It is particularly useful when the posterior distribution of a model’s parameters is difficult or impossible to calculate analytically [88]. Only six (11.5%) studies employed Integrated Nested Laplace Approximation (INLA) for model fitting and inference [30,34,39,53,56,67]. Unlike MCMC techniques, INLA performs Bayesian analyses using numerical integration and is not very flexible in modifying complex models and addressing challenges. The current literature suggests that INLA has a lower computational burden than MCMC [89]. The commonly used spatial models in this review were Bayesian hierarchical models with structured and unstructured random effects in the modelling component [90]. Unlike MCMC, INLA can perform Bayesian studies without requiring posterior sampling techniques.

The number of publications related to Bayesian spatial and spatio-temporal conditional autoregressive models showed an upward trend from 2012 to 2020, but it decreased in 2021 and 2022 (articles up to September 2022 were considered for this review). This decline could be related to the impact of COVID-19 through potentially difficulties of accessing data due to disruptions to the healthcare systems in many parts of the world. In addition, other newly evolving methodologies (e.g., machine learning) may become more common, and result in a reduction of statistical methodology [91].

A total of 26 countries had published articles on Bayesian spatial and spatio-temporal modelling in this review. Australia had 11 published articles, which is a substantial proportion of the total. The reasons for this could be the availability of good quality spatial maps and data in Australia and freely available datasets for mapping and geospatial analysis. Australia is also a very large country and diverse geographically [13] and policy makers have a significant interest in the relationships between social and SES status and health outcomes [92].

Out of 52 studies, 50 (96.2%) used DIC as a criterion to compare model goodness-of-fit, while only 2 studies used WAIC. Both DIC and WAIC employ the model likelihood function and model complexity term to compare models. The majority of studies in this review used DIC, possibly because it can be easily calculated from the samples generated by a Markov Chain Monte Carlo simulation [93].

The Poisson-based modelling approach was used in the majority of studies, specifically 30 out of 52 (57.7%). This due to the fact that Poisson distribution is specifically designed to model count data, which represents the number of occurrences of an event within a fixed unit of time and space. On the other hand, 16 studies (30.7%) used the Binomial/Bernoulli modelling approach. This can be attributed to the fact that the Poisson model is commonly used for count or discrete data with non-negative integers that are aggregated at the area level [94].

The included studies in this review used different types of covariates, like demographic socio-economic, climatology, environmental and clinical variables. Of 52 studies, 21 (41.1%) included demographic covariates such as sex, age, and race, and 19 (37.2%) of them included socio-economic variables, such as wealth index. Age, sex, and other demographic, and socio-economic variables are determinants of a multitude of health-related and other outcomes. The impact of these listed variables and other common risk factors are estimated using models.

Of the studies, 19 (36.5%) assigned flat prior distribution for the intercept [31,38,39,40,43,44,50,51,57,60,69,70,71,72,73,74,77,80,81], with 17 (32.69%) studies not reporting the prior distribution for the intercept [30,33,34,36,47,53,56,58,59,61,63,64,65,66,68,75,79]. For regression coefficients, 36 (69.23%) studies used normal distribution for prior distribution, and 11 (21.1%) articles did not report what types of prior distribution were used for the regression coefficient. For the spatial structured and unstructured random component of the model, 48 (92.3%) studies reported prior distribution of a precision term. Of these, 31 (59.6%) studies reported the spatial scale used for mapping and the distribution for parameters. Many of the included studies in this review used non-informative priors, which may be due to prior information not being available and allowing data to influence the posterior distribution [95].

Of the studies reviewed, only 8 (15.38%) studies [41,42,51,54,55,63,71,78] conducted sensitivity analysis for hyperprior distribution of precision, indicating a low rate of such analysis. The majority of studies did not include sensitivity analysis for the hyperprior distribution of precision component. Conducting sensitivity analysis for hyperprior distribution identifies influential hyperpriors and assesses the robustness of the conclusions [96]. Conducting sensitivity analysis for hyperprior distribution can inform researchers about how the choice of parameters materially affects their results. Depaoli et al. [97] emphasized the significance of prior distributions and sensitivity analysis in Bayesian methods and disease mapping. One potential reason why scholars do not conduct sensitivity analysis might be due to a lack of statistical skills or an insufficient understanding of the importance of sensitivity analysis.

Different types of adjacency matrices were used in our reviewed studies. A total of 41 (75%) of the articles used Queen contiguity adjacency matrix, 2 studies used distance-based neighbourhood matrix, and 1 study used Rook contiguity in their modelling for smoothing spatially structured random effects. Eight articles did not report what types of adjacency matrices were used in their modelling. Studies by Duncan et al. [19] noted that the specification of spatial weight matrix has a significant effect on model fit and parameter estimation, and studies by Earnest et al. [13] note that the types of neighbours specified had considerable difference in smoothing of the CAR model. This paper suggested that using a covariate as a measure of neighbourhood improves the performance of the CAR model, but it then becomes impossible to use this covariate as a risk factor, so it is important to understand the goal of the analysis. Earnest also suggested for future researchers that including spatial smoothing based on additional covariates may improve CAR model performance in disease-mapping studies.

Modifiable Area Unit Problem (MAUP) is the main source of bias in spatial and spatio-temporal modelling. MUAP affects the result when individual-level variables are aggregated at the area level. From the 52 articles, only 6 (11.5%) [55,63,66,70,73,79] studies reported reasons for using the respective spatial scale. This is likely due to most health data/surveys only collecting the postcode of patients, not the actual address, which would allow for more granular modelling.

The number of spatial units in the reviewed article ranged from a minimum of 10 and a maximum of 1235. The average spatial scale was 169 and the median was 64. The sample size in each study also varied and ranged from a minimum of 705 to a maximum of 4.05 million.

From a total of 52 studies, only 8 studies discuss methodological gaps and future research directions [40,41,51,56,65,71,74,75]. The studies conducted by Darikwa et al. [56] noted that large geographical units of analysis may mask some information of interest and be heterogeneous in outcomes and risk factors, and efficiency may be improved by having smaller units of analysis. Another study, conducted by Akter et al. [40], noted that the impact of using different smoothing priors on inference was not investigated in their study, and recommended this for future research. 

Of the studies, 44 (84.6%) reported the maps for displaying estimates geographically. Visualization of estimates geographically is a useful tool for policy and planning purposes, particularly to assist decision-making for allocation of health resources.

Of the 52 studies, only 8 studies discuss methodological gaps and future research directions [40,41,51,56,65,71,74,75].

### 4.1. Strengths and Limitations of the Review

The strength of this systematic review is that we followed the PRIMA guidelines in terms of high-quality reporting and a quality checklist for assessment of bias. A comprehensive total of seven electronic databases, with the help of a professional librarian, were searched to retrieve studies. Two authors (ZTT and GAT) searched and extracted the data of all the included studies independently using Covidence software. The limitations of this review are that there was significant variation in study methodology and included covariates, which precluded us undertaking meta-analysis, and that only studies published in English language were included. Finally, publication bias cannot be entirely avoided.

### 4.2. Implications of the Study and Its Contributions

This systematic review focusses on quantifying the problem which can lead to addressing the limitations and variability in Bayesian spatial and spatio-temporal modelling in health and health-related research. We have found a need for consistent guidelines on sensitivity analysis for choice of priors, reporting the rationale for selection of spatial units, and a clear specification of choice of neighbourhood adjacency, to better standardize and compare future publications.

## 5. Conclusions

This review highlights existing variation and limitations in the specification areal unit, adjacency matrix, and priors on Bayesian spatial and spatio-temporal models used in health research. We found that studies often fail:✓To report the rationale for the choice of areal units.✓To perform sensitivity analyses on priors.✓To evaluate the choice of neighbourhood adjacency.

These issues are important because they can potentially affect the validity and reliability of findings in health research. For example, the choice of areal units can influence the measurement of exposures and outcomes and their relationship, and the choice of neighbourhood adjacency can affect the spatial autocorrelation structure of the data, affecting the level of smoothing undertaken by the model. Similarly, the choice of priors can have a significant impact on the posterior distribution of model parameters, and sensitivity analyses can help to assess the robustness of findings to different prior specifications. A framework and heightened awareness from researchers is required for the design, analysis, and reporting of disease-mapping studies to ensure that findings are valid, reliable, and generalizable.

## Figures and Tables

**Figure 1 ijerph-20-06277-f001:**
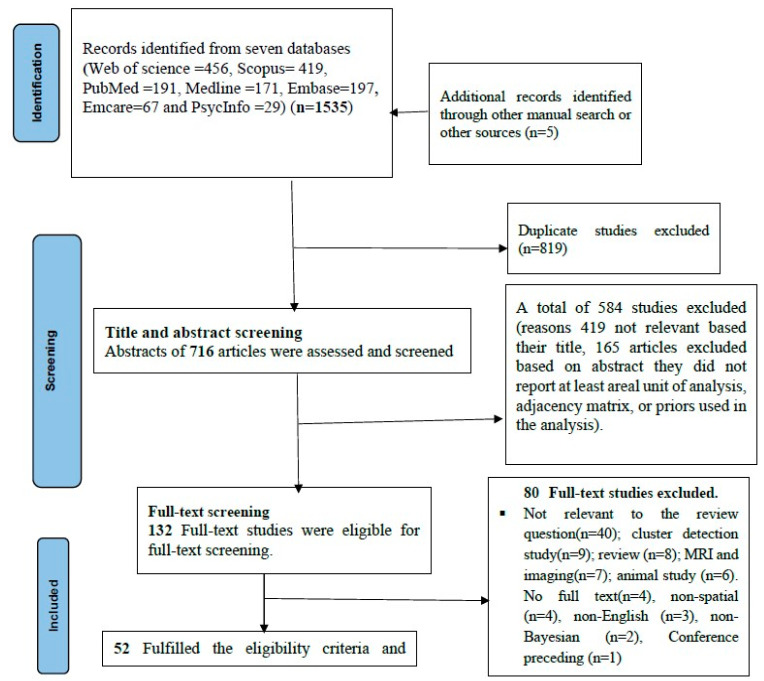
Flow chart of selection of studies for the systematic review using PRISMA checklists from 2012–2022.

**Figure 2 ijerph-20-06277-f002:**
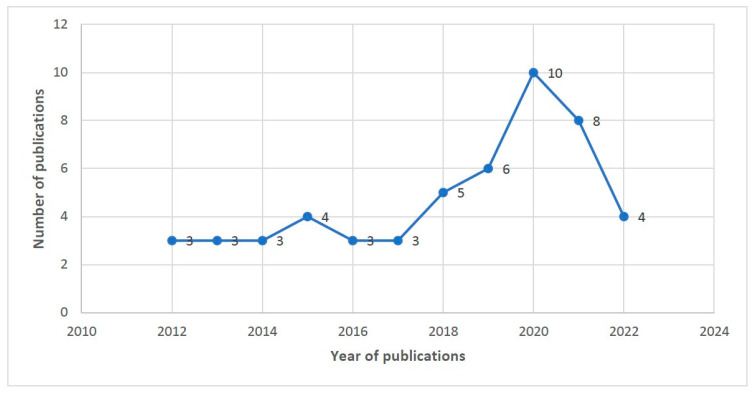
Number of studies based on year of publications from 2012 to 2022.

**Figure 3 ijerph-20-06277-f003:**
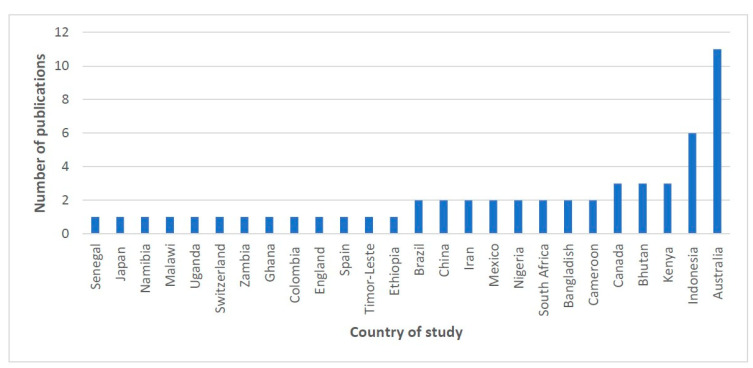
Number of included articles by country of study.

**Table 1 ijerph-20-06277-t001:** Summary of characteristics of included studies.

Characteristics	Frequency	Percentage (%)
Study category (*n* = 52)		
Infectious diseases	26	50.0
Chronic diseases	10	19.2
Maternal and child health outcome	8	15.3
Nutrition	3	5.7
Others *	5	9.6
Data source		
Survey	24	46.1
Hospital records/Administrative	20	38.4
Registry	7	13.4
Others **	1	0.03
Study design(*n* = 52)		
Ecological	42	80.7
Cross-sectional	10	19.2
Spatial unit (*n* = 52)		
District	9	17.3
County	7	13.1
Region	6	11.5
Province	6	11.5
Postcode	4	7.6
SLAs	3	5.7
LGA	2	3.8
Municipalities	2	3.8
Others ***	13	25.0
Temporal unit		
Year	42	80.7
Month	9	17.3
Week	1	0
Estimation technique (*n* = 52)		
MCMC	46	88.4
INLA	6	11.5
Software used for analysis		
WinBuGs	31	59.6
R	11	21.1
ArcGIS	8	15.3
Geoda	2	0.03
Types of distribution for outcome variable		
Poisson	30	57.7
Negative Binomial/ZIP	6	11.6
Binomial/Bernoulli	16	30.7
Model comparison		
DIC	50	96.2
WAIC	2	3.8
Effect measures reported		
RR	31	59.6
Mean	12	23.0
OR	9	17.4
Types of spatial unit		
Area	42	80.7
Point	10	19.3
Spatial structure		
ICAR/CAR/MCAR	39	75.0
GMRF	3	5.7
Not reported	10	19.2
Spatial modes used		
Bayesian spatial Poisson regression	30	57.7
Negative Binomial model	6	11.5
Binomial/Bernoulli model	16	30.7
Covariates		
Demographic	21	41.1
Socio-economic	19	37.2
Climatology/Environmental	16	30.7
Clinical	3	5.7
Others ****	7	13.4
Model diagnostic reported		
Yes	24	46.1
No	28	53.8
Map reported		
Yes	44	84.6
No	8	15.3
Script provided		
Yes	5	9.6
No	47	90.4
Burn-in, thinning, and iteration reported		
Yes	28	53.8
No	24	46.2
Reasons of uses of specified prior distribution provided		
Yes	7	13.4
No	45	85.6
Reasons for use of specified adjacency matrix provided		
Yes	2	3.8
No	50	96.2

Others *: nontuberculous mycobacteria (NTM), pertussis infection, Clostridium difficile, injury, suicide; others **: TuBerculose WEB; others ***: prefecture, sub-district, cluster, enumeration area, sub-county, SA3, census division, state; others ****: social vulnerability, distance, drinking water.

**Table 2 ijerph-20-06277-t002:** Summary of prior, sensitivity analysis, and adjacency selection in the included studies.

Id	Prior Distribution				Sensitivity Analysis for Hyperpriors		Adjacency			
	Intercept	Regression Coefficient	Spatially Structured Random Effect (Precision)	Spatially Unstructured Random Effect (Precision)	Yes	No	Queen Contiguity	Distance-Based Matrix	Rook Contiguity	Not Reported
1	Diffuse	Highly dispersed normal prior distributions	Non-informative gamma	Non-informative gamma		✓	✓			
2	Flat	Normal	Non-informative gamma (0.001, 0.001)	Non-informative gamma (0.001, 0.001)		✓	✓			
3	Flat	Normal	Gamma	Beta		✓		✓		
4	Dflat	norm (0,0.000001 )	non-informative gamma (0.001, 0.001)	Non-informative gamma (0.001, 0.001)		✓				✓
5	Flat	Normal	Non-informative gamma (0.05, 0.0005)	Non-informative gamma (0.05, 0.0005)	✓		✓			
6	NA	Normal	Non-informative gamma	Non-informative gamma		✓				✓
7	Uniform	Normal	Inverse gamma (1, 0.01)	Inverse gamma (1, 0.01)	✓		✓			
8	Uniform	Gaussian	Inverse gamma (1, 0.01)	Inverse gamma (1, 0.01)		✓	✓			
9	Normal	Normal	Inverse gamma (1, 0.01)	Inverse gamma (1, 0.01)	✓		✓			
10	NA	NA	NA	NA		✓	✓			
11	Flat	Normal with mean = 0 and 1/variance = 1 × 10^−4^	Inverse gamma (0.5, 0.005)	Inverse gamma (0.5, 0.005)		✓	✓			
12	Normal	Normal	Inverse gamma (0.5, 0.0005)	Inverse gamma (0.5, 0.0005)	✓		✓			
13	NA	NA	NA	NA		✓	✓			
14	NA	NA	NA	NA		✓				✓
15	Uninformative	Normal (0, 1000)	Weakly informative hyperpriors uniform (0, 0.001)	Weakly informative hyperpriors uniform (0, 0.001)		✓	✓			
16	Flat	Normal (0, 0.0001)	Noninformative gamma (0.1, 0.1)	Noninformative gamma (0.1, 0.1)		✓	✓			
17	Flat	Normal	Noninformative gamma (0.01, 0.01)	Noninformative gamma (0.01, 0.01)						✓
18	Normal (0, 100)	Normal	Gamma (0.01, 0.01)	Gamma (0.01, 0.01)	✓					✓
19	Flat with bounds −1 and +1	Normal (0, 0.000001)	Non-informative gamma (0.001, 0.001)	Non-informative gamma (0.001, 0.001)		✓	✓			
20	Uninformed	Uninformed	Uninformed	Uninformed		✓				✓
21	Diffuse	Normal (0, 0.0001)	Gamma (0.001, 0.001)	Gamma (0.001, 0.001)	✓		✓			
22	NA	NA	Gamma	Gamma			✓			
23	NA	NA	Gamma (0.01, 0.01)	Gamma (0.01, 0.01)		✓	✓			
23	NA	NA	Inverse gamma	Inverse gamma		✓	✓			
24	Flat	Uniform	Inverse gamma	Inverse gamma		✓	✓			
25	NA	Normal (0, 1000)	Uniform prior on (0, 1000)	Uniform prior on (0, 1000)		✓		✓		
26	NA	Informative Gaussian	Informative Gaussian	Informative Gaussian			✓			✓
27	NA	NA	Gamma	Gamma		✓	✓			
28	Dflat()	Normal	Inverse gamma (0.5, 0.0005)	Inverse gamma (0.5, 0.0005)	✓		✓			
29	NA	NA	Normal	Inverse gamma	✓		✓			
30	NA	Normal	Inverse gamma	Inverse gamma		✓				
31	NA	NA	Non-informative gamma	Non-informative gamma		✓	✓			
32	Improper uniform	Vague normal prior	Improper uniform	Normal distribution		✓			✓	
33	NA	Normal	Gamma	Gamma		✓	✓			
34	Flat	Weakly informative	Inverse gamma (0.0001, 0.0001)	Inverse gamma (0.0001, 0.0001)		✓	✓			
35	Flat	Normal	Gamma (0.5, 0.0005)	Gamma (0.5, 0.0005)		✓				✓
36	NA	Normal	Gamma	Gamma		✓	✓			
37	Normal	Normal	Inverse gamma	Inverse gamma		✓	✓			
38	Normal	Normal	Inverse gamma	Inverse gamma		✓	✓			
39	NA	Normal	Uniform prior (0, 0.001)	Uniform (0, 0.001)		✓	✓			
40	Dflat()	Normal	Gamma (0.5, 0.0005)	Gamma (0.01, 0.01)		✓	✓			
41	Flat	Normal	Weekly informative gamma	Weekly informative gamma		✓	✓			
42	Non-informative	Normal	Gamma (0.01, 0.01)	Gamma (0.01, 0.01)		✓	✓			
43	Non-informative	Normal	Gamma (0.5, 0.0005)	Gamma (0.5, 0.0005)		✓	✓			
44	NA	NA	Vague prior distribution gamma (0.5, 0.0005)	Vague prior distribution Gamma (0.5, 0.0005)		✓	✓			
45	Uniform	Normal	Gamma (0.5, 0.0005)	Gamm (0.5, 0.005)		✓	✓			
46	Flat	Non-informative normal	Non-informative gamma (0.5, 0.5)	Non-informative gamma (0.5, 0.5)		✓	✓			
47	Flat	Normal	Non-informative gamma (0.001, 0.001)	Non-informative gamma (0.001, 0.001)		✓	✓			
48	Flat	Normal	Non-informative gamma (0.01, 0.01)	Non-informative gamma (0.01, 0.01)		✓	✓			
49	Flat	Normal	Non-informative gamma (0.001, 0.001)	Non-informative gamma (0.001, 0.001)		✓	✓			
50	Flat	Normal	Non-informative gamma (0.01, 0.01)	Non-informative gamma (0.01, 0.01)		✓	✓			
51	Flat	Normal	Non-informative gamma (0.01, 0.01)	Non-informative gamma (0.01, 0.01)		✓	✓			
52	NA	NA	NA	NA		✓				✓
Total					8	44	41	2	1	8
%					15.3%	88.4%	75.0%	3.8%	1.9%	15.3%

**Table 3 ijerph-20-06277-t003:** The spatial scales used, available spatial scales, and reasons for using the spatial scale.

Article	Country of Study	Spatial Scale Used for This Study	Available Spatial Scale in the Country	Reasons for Using Spatial Scale
Adeyemi et al., 2019 [67]	Burkina Faso and Mozambique	Region for Burkina Faso Province for Mozambique	Region, province, and department (Burkina Faso) and province, district, and pesto (Mozambique)	Not mentioned
Akter et al., 2021 [40]	Australia	Statistical Local Area (SLA)	Postcode, SA1, SA2, SA3, SA4, and LGA	Not mentioned
Alam et al., 2019 [74]	Bangladesh	District	Division, district, thanas, and union	Not mentioned
Alene et al., 2021 [50]	China	Province	Province(level 1), prefecture(level 2)	Not mentioned
Amsalu et al., 2019 [51]	China	Province	Province(level 1), prefecture(level 2)	Not mentioned
Aragonés et al., 2013 [53]	Spain	Province	Autonomous community(level 1), province (level 2), and municipality(level 3)	Not mentioned
Aswi et al., 2020 [41]	Indonesia	Province	Province, district, sub-district, and village	Not mentioned
Aswi et al., 2020 [45]	Indonesia	Province	Province, district, sub-district, and village	Not mentioned
Baker et al., 2017 [54]	Australia	SLA	Postcode, SA1, SA2, SA3, SA4, and LGA	Not mentioned
Blain et al., 2013 [75]	England	County		
Chou et al., 2014 [70]	Australia	Postcode	SA1, SA2, SA3, SA4, SLA, and LGA	Postcodes are widely used by researchers in Australia because they are readily available in datasets; others not found
Cramb et al., 2015 [55]	Australia	SLAs	Postcode, SA1, SA2, SA3, SA4, and LGA	To overcome limitations of changing geographical boundaries
Danwang et al., 2021 [30]	Cameroon	Region	Region(level 1), department(level 2), and arrondissement(level 3)	Not mentioned
Darikwa et al., 2020 [56]	South Africa	District	Province, district, and municipality	Not mentioned
Desjardins et al., 2020 [76]	Colombia	Municipality	Department, municipality	Not mentioned
Dhewantara et al., 2019 [77]	Indonesia	County	Province, district, sub-district, and village	Not mentioned
Donkor et al., 2021 [31]	Ghana	Region	Region, district, and municipality	Not mentioned
Feng et al., 2015 [78]	Canada	County	Province, census division, and census sub-division	Not mentioned
Gelaw et al., 2019 [46]	Ethiopia	District	Region, zone, district, and kebele	Not mentioned
Hanandita et al., 2016 [32]	Indonesia	County	Province, district, sub-district, and village	Not mentioned
Hu et al., 2012 [42]	Australia	LGA	Postcode, SA1, SA2, SA3, SA4, and SLA	Not mentioned
Huang et al., 2017 [79]	Australia	Postcode	SA1, SA2, SA3, SA4, LGA, and SLA	Since postcode is the most readily available spatial unit in the dataset and other spatial units were not available
Ibeji et al., 2022 [33]	Nigeria	State	State(level 1), local government area (level 2), and district(level 3)	Not mentioned
Jurgens et al., 2013 [57]	Switzerland	Cantons	Cantons, district, and postcode	Not mentioned
Kandhasamy et al., 2017 [47]	India	State	State, union territories, and district	Not mentioned
Kigozi et al., 2020 [34]	Uganda	County	Region, district, county, and sub-country	Not mentioned
Lal et al., 2020 [58]	Australia	Postcode	SA1, SA2, SA3, SA4, LGA, and SLA	Not mentioned
Law, 2016 [71]	Canada	Census division	Province, census division, and census sub-division	Not mentioned
Li et al., 2020 [63]	Australia	SA3	Postcode, SA1, SA2, SA4, LGA, and SLA	Since the objective of the study was mapping at regional-level estimate
Lubinda et al., 2021 [35]	Zambia	Province	Province and district	Not mentioned
Lome-Hurtado et al., 2021 [64]	Mexico	Municipalities	Estado and municipality	Not mentioned
Lome-Hurtado et al., 2021 [65]	Mexico	Municipalities	Estado and municipality	Not mentioned
Ngwira, 2022 [68]	Malawi	District	Region, district, and traditional authority area	Not mentioned
Ntirampeb et al., 2018 [72]	Namibia	Region	Region and constituency	Not mentioned
Odhiambo et al., 2020 [69]	Kenya	Sub-county	County, sub-county	Not mentioned
Ogunsakin et al., 2022 [59]	South Africa	Region	Province, district, and municipality	Not mentioned
Okango et al., 2015 [48]	Kenya	County	County, sub-county	Not mentioned
Okango et al., 2016 [49]	Kenya	County	County, sub-county	Not mentioned
Okunlola et al., 2021 [36]	Nigeria	Enumeration Area (Cluster)	State(level 1), local government area (level 2), and district(level-3)	Not mentioned
Qi et al., 2014 [73]	Australia	LGA	Postcode, SA1, SA2, SA3, SA4, and SLA	Each LGA contains one or more SLAs and LGA boundaries are more stable compared with SLA boundaries
Raei et al.,2018 [60]	Iran	Province	Province and district	Not mentioned
Reid et al., 2012 [37]	Bangladesh	District	Division, district, thanas, and union	Not mentioned
Roza et al., 2012 [52]	Brazil	County	Region, federal unit, and municipality	Not mentioned
Saijo et al., 2018 [61]	Japan	Prefecture	Region, prefecture, and municipality	Not mentioned
Sharafi et al., 2018 [62]	Iran	District	Province and district	Not mentioned
Thiam et al., 2019 [80]	Senegal	Region	Region, department, and arrondissement	Not mentioned
Tsheten el al, 2020 [43]	Bhutan	Sub-district	District, block	Not mentioned
Wangdi et al., 2017 [81]	Bhutan	District	District, block	Not mentioned
Wangdi et al., 2018 [44]	Timor-Leste	District	Department and municipality	Not mentioned
Wangdi et al., 2022 [38]	Brazil	Region	Region, federal unit, and municipality	Not mentioned
Wangdi et al., 2020 [39]	Bhutan	District	District, block	Not mentioned
Xu et al., 2015 [66]	Australia	Postcode	SA1, SA2, SA3, SA4, LGA, and SLA	Since the only available spatial unit in the dataset is postcode

**Table 4 ijerph-20-06277-t004:** Summary of purpose of using specified prior distribution, adjacency matrices, modifiable area unit problem and modelling gaps identified.

Item	Number	%	Reference
I Reasons of use of specified prior distribution and sensitivity analysis			
✓ Non-informative prior distribution was used to minimise the risk of substantive influence on the estimates produced	8	15.3	[40,41,50,53,54,62,70,77]
✓ The values of the hyperpriors distribution are selected in a way so that our Bayesian inferences are robust and not sensitive to these choices’ different priors			
✓ Sensitivity analysis using assigned parameters of variance component done to yield a robust finding of the posterior estimate			
✓ Weakly specified hyperpriors used following a normal and inverse gamma distribution, respectively, for structured and unstructured random effect and our final choice made based on goodness of fit, computation time and plausibility of estimates			
✓ Sensitivity analysis of different assigned priors were done to see the effect of different priors on posterior estimation using DIC and models with the smallest DIC were reported and discussed.			
✓ Non-informative priors were used to minimize the risk of major influence of estimates.			
✓ Sensitivity analyses for investigating the choice of priors were carried out. The hyperprior distribution of the variance components is set to be vague to obtain most of the information from the data. The prior for the precision of the random effects (σ^2^) is often specified as a gamma distribution with scale and shape parameters both equal to 0.001. To investigate the influence of the hyperprior specifications gamma priors like gamma (0.001,0.001) and gamma (0.05,0.0005) and Uni (0,100) were done. The resulting posterior inference remains robust is selected.			
✓ For the structured and unstructured random effect precision gamma (0.5,0.0005) was specified and hyperprior is vague allows the model to get the most information from the data. Gamma (0.001,0.001) was tested for precision and the estimate remains robust and selected.			
✓ Articles did not report the reason for using specified prior distribution and not done sensitivity analysis.	44	84.6	
II Reasons of use of specified adjacency matrices			
✓ More complex specifications of the adjacency matrix, for instance using a distance-based measure introduce model complexity, unrealistic spatial dependence, and do not necessarily lead to better inference	2	3.8	[62,64]
✓ Using Queens method of adjacency for smoothing relative risk have a little impact on modelling results and it’s not common to face irregular shape.			
✓ Articles the did not mentioned reasons for use of specified adjacency matrices in their model.	50	96.1	
III Reasons of use of specified spatial scale (modifiable area unit problem)			
✓ Each LGA contains one or more SLAs and LGA boundaries are more stable compared with SLA boundaries.	6	11.5	[54,62,69,70,72,78]
✓ Since the only available spatial unit in the dataset is postcode due to that postcode used as a spatial scale for this study.			
✓ Since the objective of the study was mapping at regional level estimate we used large spatial scale due to this LGA used as spatial scale for this study			
✓ Since postcode is the readily available spatial unit in the dataset due to that postcode used as a spatial scale for this study.			
✓ To overcoming limitations of changing geographical boundaries for this study LGA used as spatial scale.			
✓ Since postcode are readily available in the dataset postcode used as a spatial scale for this study.			
✓ Articles do not reason out of use of specified spatial scale (modifiable area unit problem)	46	88.4	
IV Methodological gaps			
✓ The result of this study is analysed at the district level, which is the level at which primary health is provided in South Africa. Aggregation of the results has the effect of introducing ecological fallacy and large geographical units of analysis may mask some information of interest. Results and efficiency may be improved by having smaller units of analysis	8	15.3	[55]
✓ We used covariates at the municipality level, but this potentially masks important variations within municipalities, and to obtain more reliable results on the role of covariates as individual risk factors for overall LBW risk, it would be better to analyse birth records at the small-scale level and individual level (MAUP)			[64]
✓ Having daily or weekly data would have been preferable to examine climatic influences of dengue case as this study used monthly cases of dengue			[40]
✓ The impact of using different smoothing priors has not been done in this study and needs to be further investigated.			[39]
✓ Further studies are required to quantify the of extent the spatial differences in risk represent differences in health behaviour or in disease risks			[74]
✓ For Further study, precise estimate at local level is recommended for internal policy making and implementation			[73]
✓ First, while the level of analysis was by province/region (large geographic area), it might be desirable to examine even smaller geographical units, such as districts and cities.			[50]
✓ Explores approaches that allow the data to inform on the appropriate spatial weights to be specified is recommended.			[70]
✓ Articles did not report the methodological gaps	44	44.6	

## Data Availability

This study is a systematic review of already published articles and the extracted data are provided as a Appendix A include this statement on the section.

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
