# Peer review of "A Systematic Review of Areal Units and Adjacency Used in Bayesian Spatial and Spatio-Temporal Conditional Autoregressive Models in Health Research"

_ijerph, 2023, doi:10.3390/ijerph20136277_

Round 1

Reviewer 1 Report

Dear Authors,

Thank you for submitting this manuscript. I think the paper is quite interesting and it is worth publishing. I would like to suggest some points to the Authors:

1. In the beginning, the abstract should include a short statement on the current research gap and the reasons to show why this review is unique and worthy of publication.

2. Please add more references in the Introduction section, lines 49 - 55.

3. In the section Results, the tables are not well visible, due to bad formatting, please make corrections.

4. In the Discussion section, lines 293 – 301: Please add more references.

5. In the Conclusion section, please summarize the significant findings in bullets for more clarity.

Overall, I think the article needs just light corrections and will be ready for publication. Good luck!

Author Response

Reviewer #1

Dear Authors,

  1. In the beginning, the abstract should include a short statement on the current research gap and the reasons to show why this review is unique and worthy of publication.

Author’s Response:  Thank you reviewer for the comment. We have now added the research gap (see the revised manuscript line 34). In addition, we have provided more details in the introduction and discussion sections line 131-139

  1. Please add more references in the Introduction section, lines 49 - 55.

Author’s Response:  We have added more references for lines 49-55 see the revised manuscript (lines 96-103).  

  1. In the section Results, the tables are not well visible, due to bad formatting, please make corrections.

Author’s Response: We thank the reviewer for highlighting this. We have modified the tables accordingly (see the revised tables (1- 4))

  1. In the Discussion section, lines 293 – 301: Please add more references.

Author’s Response:  We have now cited more references in the specified lines of discussion section (see the revised manuscript line 340 and 343)

  1. In the Conclusion section, please summarize the significant findings in bullets for more clarity

Author’s Response:  Thank you. We have included main finding in bullets based on your comments (see revised manuscript line 489-491).

6.. Overall, I think the article needs just light corrections and will be ready for publication. Good luck!

Author's response: Thank you, reviewer.

Reviewer 2 Report

The authors have performed a systematic review to examine the utilization of Bayesian spatial and spatio-temporal conditional autoregressive models in health and health-related research.

Strengths:

  1. The manuscript is well-organized and comprehensive. It offers a thorough review of 52 studies utilizing Bayesian spatial and spatio-temporal conditional autoregressive models.
  2. The evaluation of studies based on quality assessment provides valuable insights into the research trends and the methodological rigor involved in the examined studies.
  3. The authors' effort to identify and describe methodological gaps and future directions in a subset of studies is commendable. This provides a pathway for further research in this domain.

However, there are several areas that could benefit from revision or additional information:

  1. A clearer explanation of the technical terms and processes associated with Bayesian spatial and spatio-temporal modeling could make the paper more accessible to a wider audience.
  2. The authors mention a decline in the number of publications related to the topic in 2021 and 2022 without providing a potential reason for this. Further exploration or discussion around this trend could enhance the manuscript.
  3. The manuscript could benefit from a deeper examination of the reasons for the prevalent use of certain techniques (like Markov Chain Monte Carlo and the Poisson-based modeling approach) over others.
  4. The manuscript noted that only a small number of studies conducted sensitivity analyses for hyperprior distribution of precision. The authors should emphasize the importance of this analysis more strongly, giving examples of potential implications when it's overlooked.
  5. The article presents a significant amount of data. However, it lacks a clear presentation of the main findings. Summarizing key findings in a more concise manner would greatly improve readability and understanding.

The quality of English in this article is generally good. The article presents complex research concepts and findings clearly and effectively. It maintains a formal, academic tone throughout, making it appropriate for a scholarly audience. The manuscript utilizes subject-specific vocabulary and terms correctly, and the sentence structure is largely correct, enhancing readability and comprehension.

However, some improvements could be made:

  1. There are instances where more concise phrasing could enhance clarity. For instance, in the sentence: "These studies reasoned that they used a specified spatial scale due to the availability of spatial information in the dataset as well as the objectives of their study and stability of individual locations." It might be clearer to say: "These studies justified their choice of spatial scale based on data availability, study objectives, and location stability."
  2. Some sentences are quite long, making them potentially difficult to follow. Breaking them up into shorter sentences might enhance readability. For example, the sentence: "The studies conducted by Darikwa et al (53) justified that large geographical units of analysis may mask some information of interest, and efficiency may be improved by having smaller units of analysis." could be broken into two sentences for clarity.
  3. It would be helpful to ensure consistency in terminology throughout the manuscript. For instance, both "spatial scale" and "geographic scale" are used to seemingly refer to the same concept. Choosing one term and sticking to it can prevent potential confusion for the reader.
  4. It might be beneficial to clarify some specific terms or phrases within the text. For example, when discussing the "quality assessment tool for modelling studies", a brief explanation of the tool might be helpful to readers who are not familiar with it.

Author Response

Reviewer #2

Strengths:

The manuscript is well-organized and comprehensive. It offers a thorough review of 52 studies utilizing Bayesian spatial and spatio-temporal conditional autoregressive models.

The evaluation of studies based on quality assessment provides valuable insights into the research trends and the methodological rigor involved in the examined studies.

The authors' effort to identify and describe methodological gaps and future directions in a subset of studies is commendable. This provides a pathway for further research in this domain.

Author's response: Thank you, reviewer.

However, there are several areas that could benefit from revision or additional information:

  1. A clearer explanation of the technical terms and processes associated with Bayesian spatial and spatio-temporal modeling could make the paper more accessible to a wider audience.

Author’s Response: Thank you for the suggestion. We have provided the explanation for technical terms with appropriate citations in the revised manuscript, lines (123-126).

  1. The authors mention a decline in the number of publications related to the topic in 2021 and 2022 without providing a potential reason for this. Further exploration or discussion around this trend could enhance the manuscript.

Author’s Response:  We have incorporated the potential reasons in the discussion section of the revised the manuscript (see the revised manuscript line 381-384)

  1. The manuscript could benefit from a deeper examination of the reasons for the prevalent use of certain techniques (like Markov Chain Monte Carlo and the Poisson-based modeling approach) over others.

Author’s Response:  Thank you reviewer for the insightful comment. We have addressed this(see the revised manuscript line 364-367 and line 397-399), but a formal model/ method comparison is beyond the scope of this research.

  1. The manuscript noted that only a small number of studies conducted sensitivity analyses for hyperprior distribution of precision. The authors should emphasize the importance of this analysis more strongly, giving examples of potential implications when it's overlooked.

Author’s Response:  Thank you for the comment. We have discussed the importance of this point in the revised manuscript line (423-430)

  1. The article presents a significant amount of data. However, it lacks a clear presentation of the main findings. Summarizing key findings in a more concise manner would greatly improve readability and understanding.

Author’s Response: -Thank you reviewer for the comment. We accepted your comment and revised the manuscript (see the revised tables (1- 4))

  1. Comments on the Quality of English Language

The quality of English in this article is generally good. The article presents complex research concepts and findings clearly and effectively. It maintains a formal, academic tone throughout, making it appropriate for a scholarly audience. The manuscript utilizes subject-specific vocabulary and terms correctly, and the sentence structure is largely correct, enhancing readability and comprehension.

Author's response: Thank you, reviewer.

  1. However, some improvements could be made:There are instances where more concise phrasing could enhance clarity. For instance, in the sentence: "These studies reasoned that they used a specified spatial scale due to the availability of spatial information in the dataset as well as the objectives of their study and stability of individual locations." It might be clearer to say: "These studies justified their choice of spatial scale based on data availability, study objectives, and location stability."

Author’s Response:  Thank you reviewer for the comment. We have revised the manuscript accordingly (see the revised manuscript line 309-312).

  1. Some sentences are quite long, making them potentially difficult to follow. Breaking them up into shorter sentences might enhance readability. For example, the sentence: "The studies conducted by Darikwa et al (53) justified that large geographical units of analysis may mask some information of interest, and efficiency may be improved by having smaller units of analysis." could be broken into two sentences for clarity.

Author’s Response:  We have revised the manuscript accordingly (see the revised manuscript line 314-315).

  1. It would be helpful to ensure consistency in terminology throughout the manuscript. For instance, both "spatial scale" and "geographic scale" are used to seemingly refer to the same concept. Choosing one term and sticking to it can prevent potential confusion for the reader.

Author’s Response: - Thank you reviewer for the comment. We accepted your comment and used consistent terminology throughout the manuscript (see the revised manuscript)

  1. It might be beneficial to clarify some specific terms or phrases within the text. For example, when discussing the "quality assessment tool for modelling studies", a brief explanation of the tool might be helpful to readers who are not familiar with it.

Author’s Response: - We have included the citation in which quality assessment tool for modelling studies adopted for this review for readers who are not familiar with it (see the revised manuscript line 322).

Reviewer 3 Report

Overall the study is good. Tables are not clearly in the file. I would suggest to short the table or change the way to present. Its hard to understand. 

On what basis 52 articles are recruited?

Method section should be elaborate more clearly. There are some spelling and grammatical mistakes so read again.

In discussion, the studies are irrelevant so i would advice to add some relevant studies. 

English good.

Author Response

Reviewer #3

  1. Overall the study is good.

Author's response: Thank you, reviewer.

  1. Tables are not clearly in the file. I would suggest to short the table or change the way to present. Its hard to understand.

Author’s Response:  Thank you reviewer for the comment. We accepted your comment and corrected it accordingly (see the revised tables (1- 4))

  1. On what basis 52 articles are recruited?

Author’s Response: - We followed a systematic search strategy and included all relevant articles that met our eligibility criteria from seven major electronic databases. The studies we have included in our review should report at least areal unit of mapping or types of adjacency matrix used or choice of prior selection using sensitivity analysis in their model development under the Bayesian framework. These issues are important because they can potentially affect the validity and reliability of findings in health research. For example, the choice of areal units can influence the measurement of exposures and outcomes, and the choice of neighbourhood adjacency can affect the spatial autocorrelation structure of the data. Similarly, the choice of priors can have a significant impact on the posterior distribution of model parameters, and sensitivity analyses can help to assess the robustness of findings to different prior specifications. We have explained all the inclusion and exclusion criteria in the methods section line 175-184.

  1. Method section should be elaborate more clearly. There are some spelling and grammatical mistakes so read again.

Author’s Response: - Thank you reviewer for the comment. We have made the corrections accordingly (see the revised manuscript line 148-151).

  1. In discussion, the studies are irrelevant so i would advice to add some relevant studies.

Author’s Response:  Thank you reviewer for the concern. We have made extensive modifications of the discussion sections, as well as the implications of the findings. We included additional references to make our discussion stronger (see the revised manuscript).

Reviewer 4 Report

Review report

Materials and Methods

2.1 Data source and search strategy

Reviewer: The authors need to better explain the method used, in this case the PRISMA method. Especially why this method was chosen, what previous studies support this method and what reliability guarantees are met by this method.

4.1 Strengths and limitations of the review

Reviewer: The study limitations need to be reviewed. There are more constraints in this type of analysis that can lead to bias in the results, e.g. the choice of keywords for database queries. More thought needs to be given to the constraints of the study.

5. Conclusions

Reviewer: It is recommended that authors before the conclusion create a section that reinforces the implications of the study and its contributions. In addition, the conclusions of the study need further reflection, they should be even more objective and concretely address which gap(s) in the literature were answered with this study. Which contribution or contributions this study has made and how these contributions can be used. It is fundamental to emphasize the importance of doing this kind of bibliometric studies.

The writing needs only one revision, it is recommended that it be read by a native English speaker

Author Response

Reviewer #4

  1. The authors need to better explain the method used, in this case the PRISMA method. Especially why this method was chosen, what previous studies support this method and what reliability guarantees are met by this method.

Author’s Response: - We thank the reviewer for this suggestion. Based on your suggestion, we have now explained the benefit of PRISMA method in the revised manuscript line number (148-151).

  1. The study limitations need to be reviewed. There are more constraints in this type of analysis that can lead to bias in the results, e.g. the choice of keywords for database queries. More thought needs to be given to the constraints of the study.

Author’s Response: - Thanks reviewer for your insightful comment. Based on your suggestion we have revised the strengths and limitations section of the manuscript (see the revised manuscript line 468-476)

  1. It is recommended that authors before the conclusion create a section that reinforces the implications of the study and its contributions. In addition, the conclusions of the study need further reflection, they should be even more objective and concretely address which gap(s) in the literature were answered with this study. Which contribution or contributions this study has made and how these contributions can be used. It is fundamental to emphasize the importance of doing this kind of bibliometric studies.
  2. Author’s Response: - Thanks reviewer for your insightful comment. Based on your suggestion we included one paragraph before the conclusion about the implications of the study and its contributions (see the revised manuscript line 477-484)
  3. The writing needs only one revision, it is recommended that it be read by a native English speaker.

Authors response: One of our coauthor is native English speaker she proofread the manuscript and corrected grammatical errors. In addition, all authors proofread and revised the manuscript based on your suggestion (see the revised manuscript)
